# Minding the knowledge-action gap: Results from a mixed-methods study of antimicrobial use among dairy farmers in central Uganda

Anica Buckel[1,2]*, Clovice Kankya[4], Mark A. Caudell[2], Tabitha Kimani[2], Alice Namatovu[3], Lordrick Alinaitwe[4], James Natweta Baguma[4], Rogers Musiitwa[4], Methodius Tubihemukama[4], Junxia Song[1], Emmanuel Kabali[1], Jorge Pinto Ferreira[5], Jeffrey LeJeune[5]

1 Animal Production and Health Division/Joint Centre for Zoonotic Diseases and Antimicrobial Resistance, Food and Agriculture Organization of the United Nations, Rome, Italy, 2 Emergency Centre for Transboundary Animal Diseases, Food and Agriculture Organization of the United Nations, Nairobi, Kenya, 3 Emergency Centre for Transboundary Animal Diseases, Food and Agriculture Organization of the United Nations, Kampala, Uganda, 4 Department of Biosecurity, Ecosystems and Veterinary Public Health, Makerere University, Kampala, Uganda, 5 Agrifood Systems and Food Safety Division, Food and Agriculture Organization of the United Nations, Rome, Italy

* anica.buckel@fao.org

## Abstract

Antimicrobial resistance (AMR) presents a growing threat to global health and food security, accelerated by human behaviours such as suboptimal use patterns. While the negative consequences of AMR will be particularly severe in low- and middle-income countries, relatively little is known about the extent and frequency of behaviours that contribute to AMR in these regions. This mixed-methods study, which included a cross-sectional survey, examines knowledge, attitudes, and practices related to antimicrobial use (AMU) among 417 smallholder dairy farmers in Uganda's Wakiso, Kampala, and Mukono districts. We found that (1) farmers' AMU practices were associated with attitudes toward veterinarians and access barriers, with many relying on private veterinarians due to challenges accessing public animal health services; and; and (2) findings support the concept of a 'knowledge-action gap,' as AMR knowledge and belief items were weakly associated with prudent antimicrobial use and related practices in our exploratory models, These findings highlight the need to rethink the current reliance on conventional knowledge-attitude-practice (KAP) interventions alone. Instead, behaviour-centred approaches informed by theory-driven behaviour change frameworks, along with interdisciplinary collaboration, may offer more profound insight into the multi-faceted factors that shape AMR-associated practices and inform more tailored interventions. Finally, given the prominent role of private veterinary service providers as trusted points of contact for farmers, there is a clear need to foster partnerships that formally recognise and engage these actors in stewardship efforts while ensuring alignment with national policy and regulatory oversight.

**Data availability statement:** All relevant data are within the manuscript and its Supporting Information files.

**Funding:** This research was funded by a grant from the Mars, Incorporated (URL: https://www.mars.com) to the Food and Agriculture Organization of the United Nations (MTF/GL/070/MRS). The funders had no role in study design, data collection and analysis, decision to publish, or preparation of the manuscript.

**Competing interests:** The authors have declared that no competing interests exist.

## Introduction

Antimicrobial resistance (AMR) is one of the most pressing public health threats and development challenges of our time and threatens the attainment of many of the United Nations 2030 Sustainable Development Goals (SDGs), including those focused on health, food security, and poverty reduction, by undermining both human and animal health systems (xx). This study therefore investigates antimicrobial use behaviours that contribute to this global challenge among smallholder dairy farmers in Uganda.

AMR occurs when microorganisms that cause infections become resistant to the effects of antimicrobial drugs, making it difficult to treat common infections in people and animals effectively. The increasing prevalence of this evolutionary process is fuelled by anthropogenic factors [1], particularly the over- and sub-optimal use of antimicrobials.

In agriculture, antimicrobials play a crucial role beyond therapeutic use and are often considered an economic asset as they limit the financial risks and labour costs when used for prophylaxis (the preventive use of antimicrobials to treat at-risk herds or animals) and metaphylaxis (the treatment of clinically healthy animals in a group that have been exposed to an infectious agent, following the diagnosis of the disease in other animals within the same group). However, these practices have emerged as major contributors to AMR development [2]. The potential transfer of resistant bacterial strains from animals to humans through agrifood systems [3] via direct contact between humans and animals, consumption of animal products, or contaminated environmental sources is a significant public health concern.

The effects of AMR are particularly pronounced in low and middle-income countries (LICs and MICs), especially sub-Saharan Africa [4], where the selection and transmission of resistant organisms and resistance genes are exacerbated by lack of access to clean water, inadequate sanitation and hygiene, and underfunded national drug regulatory agencies [5]. Moreover, a shortage of veterinary professionals in LICs and MICs makes implementing regulations at the farm level difficult. For example, a study in five African countries found that the ratio of veterinarians to livestock is about 20 times lower than that of high-income countries such as Denmark, France, Spain, and the USA [6]. This challenge of limited professional oversight is further compounded by the structure of the livestock sector itself, which is largely composed of geographically dispersed smallholder farmers [7], with greater reliance on paraprofessionals and agrovet retailers as first-line providers and limited laboratory access adding complexity to regulatory oversight. This contrasts with the agricultural landscape in many high-income countries, where production is typically consolidated into fewer, large-scale operations supported by routine herd-health planning and regulatory oversight that is easier to implement at scale.

Uganda, classified by the World Bank as a low-income country, faces the dual challenge of a growing demand for livestock products and limited veterinary infrastructure [8,9]. The Uganda National Academy of Sciences (UNAS) conducted a situational analysis on AMR under the Global Antibiotic Resistance Partnership Uganda

[10], showing increasing trends in antimicrobial resistance. The livestock sector in Uganda faces a high burden of bacterial diseases, with treatment options being increasingly compromised by resistance to commonly used antimicrobials, exceeding 50 percent in many cases. In Uganda, the livestock sector accounts for about 17 percent of agricultural value added and 4.3 percent of GDP, with 58 percent of households dependent on livestock for their livelihoods. Most of them are subsistence-oriented smallholders [9], with cattle being the most crucial livestock sub-sector, contributing over 40 per cent to the value of livestock production and about 7 per cent to agricultural production. Cattle farming provides the population with income, food, draft power, insurance, savings, social capital and other goods and services [9].

To prevent, slow down, and control the spread of resistant organisms, Uganda developed a One Health AMR National Action Plan (NAP-AMR) in 2018. This plan underscores the need for data to inform evidence-based interventions, particularly in the livestock sector, as research in Uganda has centred mainly on AMR in humans and hospital settings [11].

The NAP-AMR specifically highlighted this data gap and the perceived lack of Knowledge, Attitudes, and Practices (KAP) studies in the livestock sector, for instance, to assess communication and education needs related to AMR. Although more KAP studies have recently emerged, for example, surveys among livestock farmers in Kyegegwa District [12] and Kassanda District [13], these remain limited in geographic scope and species coverage, underscoring the continued need for more evidence from the dairy sector. The insights generated through a KAP study are essential for identifying knowledge gaps and potential areas for educational intervention, thereby informing the development of targeted strategies and strengthening Uganda's capacity to address AMR. The findings from this research are intended to provide evidence that can inform the development of targeted policy recommendations and intervention designs aimed at promoting more prudent antimicrobial use.

Therefore, this study was conducted to (1) to assess the knowledge, attitudes, and practices related to antimicrobial use and antimicrobial resistance among smallholder dairy farmers in central Uganda, (2) to identify demographic, farm-specific, and attitudinal factors associated with antimicrobial misuse and animal health-seeking behaviours; and (3) to explore the relationship between AMR knowledge and AMU practices.

## Methods

### Study design

An interdisciplinary research team of animal health experts, epidemiologists, and social and behavioural scientists from Makerere University and the Food and Agriculture Organization of the United Nations (FAO) developed and implemented the research project. This study employed a cross-sectional, mixed-methods design, integrating both quantitative (structured questionnaire survey) and qualitative (focus group discussions). The study was implemented in two phases, a qualitative phase, followed by a quantitative phase.

### Study areas

This study was conducted in the Kampala Metropolitan Area, which encompasses the Mukono, Wakiso, and Kampala districts in central Uganda (see Fig 1).

Wakiso, which partly encircles the capital city of Kampala, is predominantly rural, with some peri-urban areas hosting a population of over 3.1 million [14]. Mukono has a mix of urban and rural areas, where small-scale agriculture, fishing, and small businesses dominate livelihoods. Its population is approximately 596,804.

### Study population and sampling strategy

The primary sampling unit was the individual dairy farmer (the person responsible for day-to-day herd management and animal health decisions) within each household. The target population consisted of smallholder dairy farmers operating within the three study districts. Smallholders were operationally defined for this study as farmers owning between 1 and 15

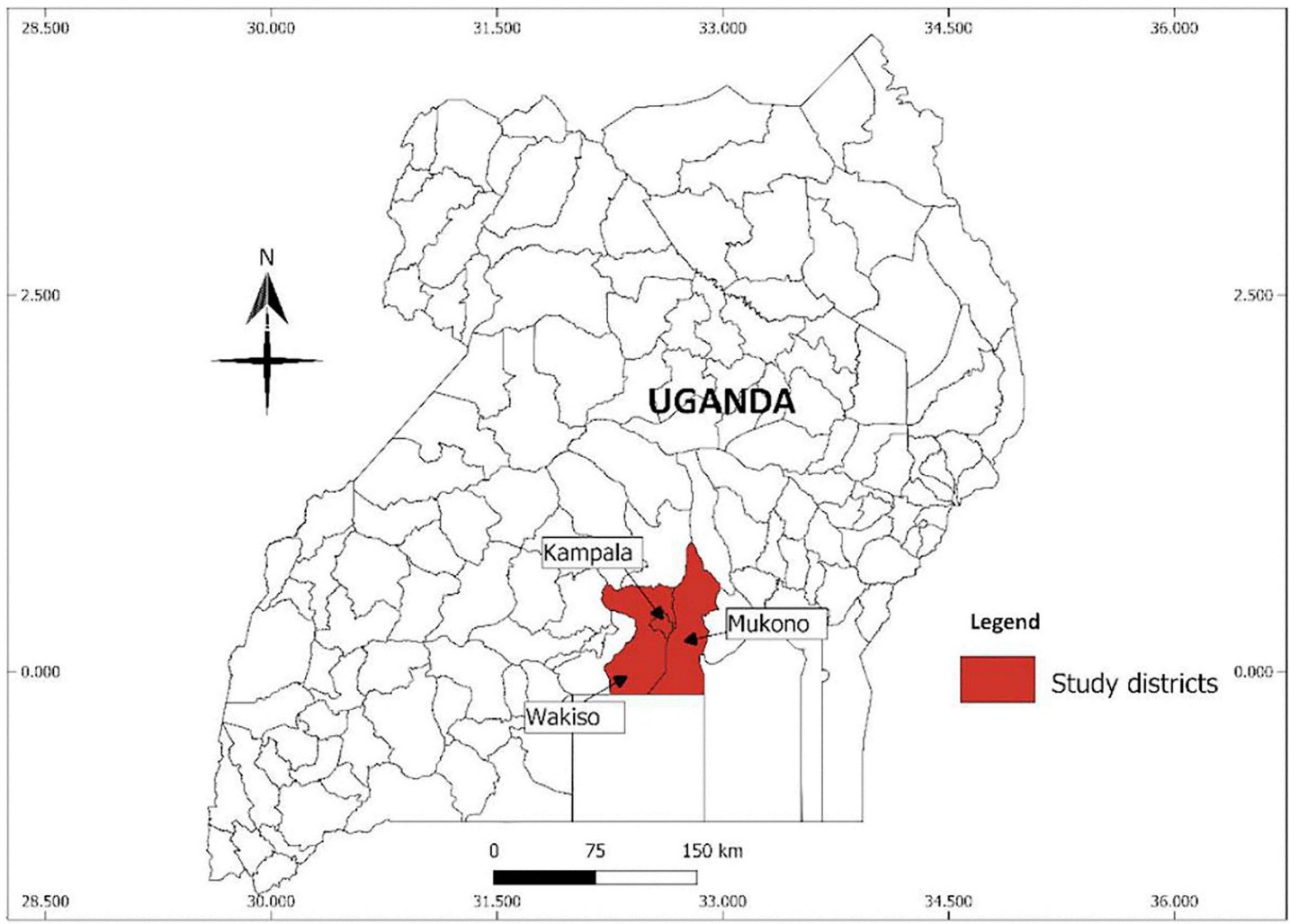

**Fig 1. Map of Uganda showing the study districts.** Source: https://open.africa.

dairy cattle. This threshold was chosen to reflect Ugandan smallholder systems, where herds are typically between 1–10 animals, with mean herd sizes ~2–4 in districts near our setting [15,16], while also accommodating regional heterogeneity in Uganda where "small-scale" systems can exhibit larger average herds [17].

The sample size was determined using an *a priori* power analysis conducted with G*Power. To detect a small-to-medium effect size ($f^2 = 0.04$) with 80% power at a significance level of $\alpha = 0.05$ for a multiple regression model with up to 10 predictors, a total sample size of 416 participants was required.

The sampling frame was obtained from the District Veterinary Office in each study district through lists of registered dairy farmers maintained by the public veterinary/extension services. From these lists, we recruited a convenience sample of eligible farmers. Government extension staff occasionally assisted enumerators with the farm location but did not select participants. No recruitment was conducted through private veterinary clinics or their client lists. The final survey sample comprised 417 farmers, one respondent with an exceptionally large herd (100 cattle) was excluded as an outlier from regression analyses, resulting in an analytical sample of 416.

Eligibility criteria for inclusion in the quantitative survey were:

- Operating a dairy farm within the selected study areas.

- Being a smallholder, operationally defined for this study as owning between 1 and 15 dairy cattle.

- Being the primary individual (aged 18 years or older) responsible for the daily management and animal health decisions on the farm.

- Willingness to participate and provide written informed consent.

Exclusion criteria were:

- Large-scale commercial dairy enterprises

- Individuals not directly involved in the day-to-day management or animal health decisions of the dairy farm.

The final sample for the survey consisted of 417 farmers. One participant with an exceptionally large herd (100 cattle) was excluded from the regression analyses as an outlier, resulting in a final analytical sample of 416.

For the qualitative phase, participants were purposively selected from the same pool of eligible participants.

## Data collection and survey development

FGDs were conducted before the main survey to inform and refine questionnaire design. Three Focus group discussions (FGDs) were conducted with farmers operating small and medium-sized dairy cattle farms. Following a semi-structured interview guide, themes discussed in the FDGs included a variety of management topics, such as antimicrobial use, disease prevention, relationship with animal health professionals (AHP) and animal health-seeking behaviours.

Following the qualitative phase, the KAP survey instrument was contextualized based on the respondents from participants during the FGDs to reflect local practices. Enumerators then received training on standardised questionnaire administration, neutral probing techniques, building rapport, and ethical conduct. The survey was first piloted by Makerere University and FAO research teams with a small group of dairy farmers (not included in the main study sample) and further refined based on feedback to minimise ambiguity and potential misinterpretation of questions.

Trained enumerators administered the final questionnaire in face-to-face interviews between 15 March and 17 April 2023 using Kobo Collect® to ensure standardised data entry and reduce errors. Surveys took between 45 minutes and 90 minutes to complete. This variation was primarily due to differences in respondent engagement, farm complexity, and potential interruptions that are expected during field settings.

## Public involvement

While the primary research question was informed by Uganda's National Action Plan on AMR, the public was involved early in the study through focus group discussions and interviews with smallholder dairy farmers. These engagements helped contextualise the study tools and refine survey questions to reflect local practices and priorities around antimicrobial use. Farmers provided input on the language, relevance, and comprehensiveness of the questions, and identified additional areas they felt were important to explore. Recruitment involved local veterinarians and peer referrals. Although farmers were not involved in conducting the study or data analysis, dissemination plans, including community feedback and co-designed interventions, were shared during consent and will be implemented in a subsequent project phase.

## Data analysis

**Statistical analysis.** Quantitative data were analyzed using StataNow 19.5. Descriptive statistics were generated, followed by LASSO variable selection and multivariable logistic regression. Associations were assessed using odds ratios

with 95% confidence intervals. Model fit was evaluated using pseudo $R^2$ and diagnostic tests. All data for the variables analysed in this study were collected directly from the participating farmers through the administration of a structured KAP survey questionnaire.

We initially conducted an a priori power analysis based on multiple linear regression assumptions. Although logistic regression was ultimately used due to the binary nature of the outcome, the key determinants of power (effect magnitude, α level, and number of predictors) are shared across both modelling frameworks. To evaluate sample adequacy for logistic regression, we assessed the events-per-variable (EPV) ratio for each model. Recent simulation studies demonstrate that models with 5–9 EPV can yield stable coefficient estimates and acceptable Type I error rates, particularly when predictors are not highly correlated and effect sizes are not extremely small [18,19]. All of our models met these conditions, and diagnostic checks indicated stable estimates, supporting the adequacy of the achieved sample size for logistic modelling.

Models were specified in two domains, including practices relevant to prudent antimicrobial use and animal health seeking practices. For prudent antimicrobial use practices, dependent variables included exclusive reliance on animal health professionals to administer antibiotics, whether the first step upon noticing sickness in cow/herd was to call an animal health professional, whether withdrawal periods were observed for milk, and whether the correct antibiotic dosage and treatment days were always followed, as reported by the interviewee. Examined correlates of animal health seeking practices included exclusive reliance on animal health professionals for animal health advice, reliance on informal sources for animal health advice, and whether challenges were experienced by the farmer in accessing quality animal health services, as reported by the participant. Table 1 provides variable definitions and descriptive statistics for the dependent variables.

To identify correlates within the two domains a two-step modeling approach was utilized. First, LASSO (least absolute shrinkage and selection operator) regressions were specified to identify a parsimonious set of correlates associated with the outcome of interest within the two domains. LASSO performs variable selection and regularization through imposing a penalty on the absolute size of regression coefficients, thereby shrinking less informative coefficients toward zero (i.e., removing from these variables from the model) [20]. Variables with non-zero coefficients in the LASSO model (selected using 10-fold cross-validation) were then entered into multivariable logistic regression models to estimate odds ratios (ORs) and 95% robust confidence intervals

Variable sets included in LASSO regressions were the same within domains and included: 1) farm-specific factors, such as herd size, experience in dairy keeping, and the reported number of diseases common to the dairy herd; 2) demographic factors that may impact husbandry practices, including age, gender, and education; and 3) knowledge and beliefs around AMR-relevant topics (e.g., knowledge of AMR, withdrawal, etc.). These variables were chosen as they represent dimensions that could inform intervention content (e.g., training to increase knowledge and awareness of AMR) or intervention design. Based on our observations within the field and analysis of qualitative data, geographic variables (i.e., sub-counties) were also included in the animal health-seeking model. Importantly, we also include a variable indicating whether a veterinarian was present at the interview as enumerators were accompanied by government veterinarians to the farms to introduce the study and enumerator. See Table 2 for variables names, definitions, and whether the variable was included in the prudent AMU domain and/or animal health seeking domain. Descriptive statistics of these variables are provided in the Results and expanded statistics (e.g., range) are provided in S1 Table in S1 Annex in the Supplementary Material.

For most LASSO models, potential correlates were selected through the tuning parameter (λ), which is identified via ten-fold cross-validation to minimize the model's mean prediction error. However, when the number of outcome events was limited, and thus model overfitting was a concern, visual inspection of the LASSO path using the *lassoknots* command in StataNow 19.5 was used to identify the "elbow" point corresponding to a parsimonious set of predictors. Visual inspection of LASSO paths was further guided by recommendations for logistic regression regarding the acceptable number of events per variable, with 5–9 events per variable used as guidance (e.g., a

**Table 1. Definitions and descriptive statistics for dependent variables. N = 417 as outlier removed from regression models included.**

| Model name | Variable Definition and categories | Freq. | Percent |
|---|---|---|---|
| Correct treater | Animal Health Professionals exclusively administer antibiotics to dairy cattle | | |
| | No | 145 | 34.77 |
| | Yes | 272 | 65.23 |
| Call Vet First | On first signs of sickness, an animal health professional is called | | |
| | No | 41 | 9.83 |
| | Yes | 376 | 90.17 |
| Observe Withdrawal | Milk from cows undergoing treatment or within withdrawal period is discarded and not sold or consumed | | |
| | No | 126 | 30.22 |
| | Yes | 291 | 69.78 |
| Correct Dosage | When antibiotics are administered, the recommended dosage is always followed (amount and number of days) | | |
| | No | 126 | 30.22 |
| | Yes | 291 | 69.78 |
| Professional AH sources | When seeking general animal health advice, qualified animal health professionals (public or private) are consulted | | |
| | No | 149 | 35.73 |
| | Yes | 268 | 64.27 |
| Informal AH sources | When seeking advice for general animal health, informal sources (friends, family, fellow farmers, internet) are consulted | | |
| | No | 232 | 55.64 |
| | Yes | 185 | 44.36 |
| Vet Challenges Experienced | Whether challenges are reported in accessing professional veterinarians or in services provided | | |
| | No | 111 | 26.62 |
| | Yes | 306 | 73.38 |

model where the dependent variable has 50 events can have 5–10 variables) [18,19]. LASSO coefficient paths and deviance plots used to identify the elbow points are provided in Table S2-S8 in S1 Annex in the Supplementary Materials. Where the number of variables selected by LASSO was larger than through visual inspection/event guidance, models including all LASSO selected variables were run (see Table S9-S10 in S1 Annex). Substantive interpretation of results between full models and models aided through visual inspection/event guidance were the same.

Regression diagnostics were performed for all logistic models specified with LASSO selected correlates. Model diagnostics indicated generally good fit across the logistic regression models. Area under the ROC curve (AUC) values ranged from 0.70 to 0.86, indicating acceptable to strong discrimination, and the percentage of correctly classified observations ranged from ~70% to ~90% (see Table S11 in S1 Annex in Supplement) The Hosmer–Lemeshow goodness-of-fit tests were non-significant for all models, suggesting no evidence of lack of fit (all $p > 0.31$) (see Table S11 in S1 Annex in Supplement). Variance Inflation Factors (VIFs) for all predictors were well below 2, indicating no meaningful multicollinearity (see Tables S13-S14 in S1 Annex in Supplement). Sensitivity analyses excluding observations with Pearson or deviance residuals > |2| or leverage > 2·k/n produced substantively similar results, indicating that findings were not driven by influential data points (see Figures S1-S21 in S1 Annex in Supplement for plots of leverage, Pearsons and deviance by predicted probabilities).

**Table 2. Variables entered into Lasso regressions for outcomes of antimicrobial use. AMU indicates antimicrobial use models and AH animal health seeking models.**

| Variable | Definition | Model |
|---|---|---|
| Age | A categorical variable of age 18–24, 25–30, 31–40, 41–50, 51–60, >60 | AMU/AH |
| Gender | 1 = Female 0 = Male | AMU/AH |
| Education | A categorical variables of no formal, primary, secondary, and tertiary | AMU/AH |
| Additional training | A binary variable (1 = Yes, 0 = No) indicating whether respondent had any training related to animal health | AMU/AH |
| Dairy>50 income | Whether income from dairy production represented 50% or more of total household income (=1) or not (=0) | AMU/AH |
| Dairy keep | A categorical variable indicating years keeping dairy Under 2 years, 2–5 years, 6–10 years, 11–20 years, 21–30 years, Over thirty years | AMU/AH |
| Grazing | Whether dairy herd was free ranging (=1) or zero-grazing (=0) | AMU/AH |
| Correct AH Source | Whether general animal health advice was usually sought from an animal health professional (=1) or friends/fellow farmers/internet (=0) | AMU |
| Correct AM Source | Where advice on the use of antimicrobials is sourced, including from an animal health professional (=1) or friends/fellow farmers/internet (=0) | AMU |
| Define AMR | Whether respondent can correctly define antimicrobial resistance "bugs" that do not respond to drugs (=1) or not (=0) | AMU/AH |
| Withdrawal Information | Whether respondent reported being told by an animal health professional about the importance of observing antibiotic withdrawal periods in milk consumption and sales | AMU |
| Misuse Information | Whether respondent had been told by an animal health professional about the dangers of drug misuse (e.g., overuse, underuse) (=1) or not (=0) | AMU/AH |
| Bigger/Faster agree | Whether respondent agreed with the statement that "Injectables" can help cattle grow bigger/faster (=1) or disagreed (=0) | AMU |
| Stop treat agree | Whether respondent agreed with the statement that "You can stop giving cow full antimicrobial course if symptoms improve" (=1) or disagreed (=0) | AMU |
| Prevent agree | Whether respondent agreed with the statement that "Giving healthy animals injectables will prevent future sickness" | AMU |
| Withdrawal agree | Whether respondent agreed with the statement that "After using veterinary drug on an animal, you should wait to use milk (=1) or disagreed (=0) | AMU |
| AHP access | Whether respondent agreed with the statement that "Animal health professionals in my area are easily accessible" (=1) or disagreed (=0) | AMU/AH |
| No farm records | Whether respondents kept records of their dairy business, including sales, treatments, inputs, etc.) (=1) or not (=0) | AMU |
| Herd size | Continuous variable of dairy cattle herd size | AMU |
| Diseases | Number of diseases reported as occurring in a respondent's dairy herd | AMU/AH |
| No-non-AHP consult | If a farmer reported to not get animal health advice from informal sources, including friends. fellow farmers, ag companies, internet | AMU/AH |
| Num-Vet challenges | Number of challenges to getting veterinary services (cost, distance, poor network, not aware of vet, unavailable) | AMU/AH |
| Satisfaction Gov. Vet | Categorical variable indicating whether respondent was satisfied (2), somewhat satisfied (1) or not satisfied(0) with govt. veterinarians services | AH |
| Satisfaction Priv. Vet | Categorical variable indicating whether respondent was satisfied (2), somewhat satisfied (1) or not satisfied(0) with private veterinarians services | AH |
| Sub-county | Categorical variable indicating the sub-county where respondent resided | AH |
| Vet Present | Whether the veterinarian (who accompanied data collectors) was present at interview | AMU/AH |

While the prudent use models included the entire analyzed dataset (N = 416), observations for two of the animal health seeking models were smaller with 402 observations. This reduction was due to LASSO retention of variables assessing satisfaction with professional veterinary services, which included an option for "no comment" and these observations were removed prior to the analysis.

## Qualitative analysis

Audio-recorded, semi-structured interviews were transcribed verbatim and, when necessary, translated into English. The transcriptions were then anonymised. Two authors (AB and MAC) independently analysed the data to reduce bias. Codes were generated inductively from the transcripts using reflexive thematic analysis [21]. A codebook was developed iteratively, combining deductive codes informed by prior research on antimicrobial use and resistance, and inductive codes emerging from the data (see Table S14 in S1 Annex in Supplement). Following coding, segments relating to each theme were compared within and across transcripts to identify patterned meanings, divergences in practice, and contextual conditions shaping antimicrobial use and disease management behaviours The free and open-source qualitative research tool *Taguette* [22] was used to generate codes.

To assess coding consistency, the three focus group discussions independently coded by AB and MAC. Overall coding agreement across all transcripts was approximately 73%, indicating strong consistency in code application. Inter-rater reliability was calculated using Cohen's κ for each code. Agreement was substantial for antimicrobial resistance-related codes (κ ≈ 0.60–0.75) and perfect for withdrawal compliance (κ = 1.00). Several low-frequency codes exhibited high percent agreement but κ values near zero; this pattern is expected due to the κ prevalence paradox, where κ is suppressed when codes occur infrequently despite coder alignment [23]. See S3 Annex for a summary of the qualitative codebook.

## Ethical approval

All participants provided written informed consent. Thumbprint signatures were requested for those unable to write. They also received an information sheet explaining the study's objectives, benefits, and risks and were assured of their right to withdraw without any negative consequences. Confidentiality and voluntarism were maintained throughout. The study received scientific and ethical clearance from the Uganda National Council for Science and Technology, with the reference No. HS 2607ES.

## Results

We first describe the demographic and socioeconomic characteristics of the sample, then present model results across the two domains, followed by qualitative findings that contextualize the quantitative patterns.

### Demographics and socio-economic characteristics of respondents entered into LASSO regressions

Participants were between the ages of 41–50 (25.18%, *n* = 105), 60% (*n* = 252) were males, and had a primary education (44%, *n* = 184). Approximately 40% (*n* = 166) of respondents reported that their primary source of income was dairy farming, with most farmers (80%, *n* = 330) having more than 6–10 years of experience in dairy farming. About a third of farmers had previous training (e.g., training from government or non-governmental organisation) related to animal production (27.58%, *n* = 115), including biosecurity, animal health, and marketing. The average dairy herd size was 4.25 cows, with a median of 3 cows. The most common system for raising dairy cattle was free range (41.97%, *n* = 175) followed by zero-grazing (39.57%, *n* = 165), tethering (28.30%, *n* = 118), and paddocking (10.07%, *n* = 42).

These characteristics are broadly consistent with the profile of smallholder dairy farmers in the study districts, with a largely male-dominated sector with comparatively good access to extension services and non-farm business opportunities linked to proximity to Kampala. Farmers are often relatively well educated and engaged in other enterprises [15,17,24]. The following findings should be interpreted within this context.

## AMR-relevant practices entered into LASSO regressions

The most common misuse practices were using antimicrobials to prevent cattle illness (21.78%, $n = 90$) and to boost appetite (41.5%, $n = 172$). Few participants (<10%) used antimicrobials to acclimatise cattle to a new environment, promote faster or more significant growth, and increase milk production. About a quarter (25.8%, $n = 98$) reported treating animals with antimicrobials themselves, with fewer saying this is done by their workers (17.8%, $n = 67$) or peers (10.4%, $n = 38$). More than half of the participants (57.5%, $n = 217$) reported that the antimicrobials they used were never or rarely prescribed by a veterinarian. See Table S

Regarding AMR-relevant knowledge and beliefs, around a quarter of participants could define AMR correctly (24.7%, $n = 103$), which meant they mentioned bacteria or microorganisms that do not respond to drugs. Fewer participants could define antibiotic residues (18.5%, $n = 77$). Most respondents reported being told about the harm of using too many medications by an AHP (68.6%, $n = 286$). In terms of agreement to following prudent practices, 45% ($n = 189$) agreed with the practice to stop an entire course of antimicrobials if symptoms improved, 65% ($n = 281$) agreed that if injectables are given too often they might stop working, and 46.6% ($n = 194$) agreed that consultation from an AHP is necessary before administering antimicrobials to cattle. Fewer respondents agreed that giving healthy cattle antimicrobials would prevent future sickness (31.3%, $n = 131$) or help cattle grow faster (25.90%, $n = 108$). Finally, about three-quarters of respondents (73.6%, $n = 307$) agreed that milk should not be consumed or sold after administering antimicrobials.

Farmers reported to access both professional and informal sources for animal health advice and treatment. Regarding professional sources, around 78% of respondents (N = 325) reported to seek advice from private veterinarians on animal health while only 26% (N = 110) reported seeking out government veterinarians. Likewise, 60% of farmers (N = 251) reported to rely on private veterinarians to administer antibiotics while just 5.5% (N = 23) reported to rely on government veterinarians. Differences in access to private and government veterinarians was reflected in satisfaction levles with 70% of farmers (N = 290) satisfied with services of private veterinarians while just 27% (N = 112) were satisfied with government veterinarians. In terms of informal sources, 24% of farmers (N = 103) responded that they rely on other farmers, family or friends for treatment advice, while 14% (N = 57) relied on attendants at agrovets shops and just 2% (N = 9) relied on internet sources. Sixteen percent (N = 66) relied on their own experiences with veterinary care to inform treatment decisions.

## Analysis of antimicrobial misuse

**Model 1: Correlates of prudent antimicrobial use and related practices.** Results of the logistic regression analysis are shown in Table 3. Age was positively associated with correct practices and always following the correct dosage. Farmers reporting more veterinary challenges had lower odds of reporting that professionals administered antibiotics and of observing withdrawal periods, although they had higher odds of consulting a professional as a first treatment step. Reliance on animal health professionals was consistently associated with improved prudent AMU outcomes, particularly calling a professional at first signs of sickness and following correct dosage and treatment days. Not seeking advice from informal sources doubled the odds of having professionals treat animals. Perceiving veterinarians as accessible was associated with higher odds of reporting the correct first treatment step. Having a veterinary professional present during the interview increased reporting that professionals administer antibiotics but was negatively associated with reporting correct dosage. Being informed about withdrawal practices significantly increased reporting of observing withdrawal periods. Reporting more common herd diseases was associated with lower odds of observing withdrawal and correct dosage. Farmers not keeping records had lower odds of reporting correct dosage. Finally, beliefs supporting preventive antibiotic use or early cessation of treatment were associated with substantially lower odds of reporting correct dosage and treatment days. Model fit (pseudo-$R^2 = 0.07$–$0.28$) indicated modest to low explanatory power.

**Table 3. Correlates of prudent AMU and AMR related practices.**

| Correlates | Correct Treater | First Treatment Step | Observe Withdrawal | Observe Correct Dosage |
|---|---|---|---|---|
| Age | 1.205* | | | 1.234* |
| | (1.029 - 1.412) | | | (1.031 - 1.477) |
| Grazing | | | | 0.634 |
| | | | | (0.369 - 1.090) |
| Diseases | 0.881 | | 0.677* | 0.780* |
| | (0.749 - 1.036) | | (0.487 - 0.941) | (0.638 - 0.954) |
| No Farm Records | | | | 0.468** |
| | | | | (0.271 - 0.809) |
| Correct_AH_source | | 8.893*** | | 2.202* |
| | | (4.018 - 19.680) | | (1.098 - 4.416) |
| No non-AHP source | 2.016** | | 0.474 | |
| | (1.271 - 3.199) | | (0.145-1.558) | |
| AHP access | 1.87 | 4.011** | | |
| | (0.940 - 3.719) | (1.491 - 10.793) | | |
| Num_Vet_challenge | 0.747* | 1.722* | | 0.627** |
| | (0.576 - 0.968) | (1.084 - 2.736) | | (0.460 - 0.856) |
| Withdrawal Info | | | 3.63** | |
| | | | (1.434 - 9.169) | |
| Prevent agree | | 0.508 | | 0.464* |
| | | (0.227 - 1.139) | | (0.259 - 0.833) |
| Stop Treat Agree | | 0.454 | | 0.466** |
| | | (0.204 - 1.012) | | (0.269 - 0.806) |
| Vet at interview | 2.722*** | 1.324 | 0.812 | 0.316*** |
| | (1.696 - 4.370) | (0.631 - 2.780) | (0.431 - 1.529) | (0.184 - 0.543) |
| Constant | 0.954 | 0.533 | 0.116 | 24.594*** |
| | (0.335 - 2.714) | (0.143 - 1.983) | (0.045-0298) | (5.801 - 104.261) |
| Pseudo R | 0.12 | 0.18 | 0.07 | 0.24 |
| Observations | 416 | 416 | 416 | 416 |

Robust 95% confidence intervals in parentheses.

*** p<0.001, ** p<0.01, * p<0.05.

## Model 2: Correlates of animal health services seeking

Results of the logistic regression analysis are presented in Table 4. Satisfaction with private veterinary services was associated with higher odds of seeking professional animal health advice. In contrast, farmers who reported using informal sources for health information had substantially lower odds of seeking professional services and higher odds of continuing to rely on informal sources. The number of commonly reported herd diseases was positively associated with seeking both professional and informal advice, suggesting that higher perceived disease burden increases overall information-seeking behavior. Additional livestock health training was also associated with increased likelihood of using informal sources. Receiving information on antimicrobial misuse, relying on professional veterinary sources, and being interviewed in the presence of a veterinarian were all associated with lower odds of seeking informal sources of advice. Reporting challenges accessing veterinary services was more

**Table 4. Correlates of seeking professional or informal sources for animal health information and challenges accessing veterinarians.**

| Correlates | Professional AH Sources | Informal AH Sources | Vet Challenges Experienced |
|---|---|---|---|
| Age | | | 0.802* |
| | | | (0.670 - 0.959) |
| Dairy>50 income | 0.665 | | 2.689*** |
| | (0.304 - 1.456) | | (1.544 - 4.685) |
| Grazing | 0.420* | | 2.135* |
| | (0.196 - 0.897) | | (1.072 - 4.253) |
| Additional training | | 1.874* | |
| | | (1.143 - 3.070) | |
| Diseases | 1.511* | 1.295** | 1.550*** |
| | (1.025 - 2.229) | (1.093 - 1.533) | (1.215 - 1.977) |
| Misuse Information | 0.363 | 0.458* | |
| | (0.117 - 1.126) | (0.247 - 0.847) | |
| Define AMR | 0.372 | 0.690 | |
| | (0.132 - 1.051) | (0.406 - 1.173) | |
| Satisfaction Priv. Vets | 1.856* | | 0.588 |
| | (1.116 - 3.088) | | (0.340 - 1.018) |
| Professional AH Source | | 0.284* | |
| | | (0.106 - 0.760) | |
| Informal AH Source | 0.244*** | | 2.270* |
| | (0.144 - 0.416) | | (1.188 - 4.336) |
| Num incorrect AH source | | | 2.213* |
| | | | (1.205 - 4.064) |
| Sub_county_3 | 0.186*** | 1.758 | 1.922 |
| | (0.078 - 0.441) | (0.749 - 4.127) | (0.662 - 5.579) |
| Sub_county_4 | | 0.385** | 0.478* |
| | | (0.189 - 0.787) | (0.231 - 0.987) |
| Sub_county_9 | | 1.882 | |
| | | (0.737 - 4.807) | |
| Sub_country 12 | 0.213** | 2.148* | 0.313 |
| | (0.069 - 0.656) | (1.034 - 4.460) | (0.079 - 1.232) |
| Sub_county_13 | | 0.323* | 7.298** |
| | | (0.134 - 0.778) | (2.037 - 26.151) |
| Vet at interview | 0.779 | 0.353*** | 1.690 |
| | (0.334 - 1.814) | (0.213 - 0.585) | (0.939 - 3.044) |
| Constant | 7.590 (1.344-4.251) | 2.900 (0.960-8.755) | 2.637 (0.663-10.487) |
| Pseudo R | 0.25 | 0.13 | 0.23 |
| Observations | 402 | 416 | 402 |

Robust 95% confidence intervals in parentheses

*** p<0.001, ** p<0.01, * p<0.05

likely among farmers whose dairy income represented a larger share of household income, those with greater disease burden, farmers with free-grazing herds, and those who relied on informal sources of advice, but was less likely among older farmers. Model fit was modest (pseudo-$R^2$ = 0.13–0.25). Full odds ratios and 95% confidence intervals are presented in Table 4.

### Insights from qualitative data

Analysis of the qualitative data provided further context for the findings from the quantitative results, particularly regarding antimicrobial use patterns. Table 5 provides contextual detail on participant characteristics and production settings,

**Participants experience treatment failure**. The FGD discussions revealed a general awareness among participants that the efficacy of treating some infections with antimicrobials was decreasing. *"There is a certain drug we used to buy in older days, and it used to work very well, but nowadays the same drug does not cure the intended disease, we wonder why"* (FGD Mukono*).*

**Varied attribution to what causes treatment failures**. Several explanations for treatment failure were identified. Overusing antimicrobials was a concern for some participants: *"Sometimes you buy what you are used to, but when you overuse it, it also ceases to work."* (FGD Wakiso). However, others offered alternative explanations, including the poor quality of commercially available drugs or delayed diagnoses: "*This resistance is mostly because the drug manufacturers have diluted the drugs. It is also because we do late diagnosis, and so the animals do not respond to treatment".* Another farmer shared a similar sentiment: "*I think the producers have different drugs with different capacities, and we access cheaper versions which are not well effective.*" (FGD Mukono)

**Reliance on AHPs for administering antimicrobials and as a primary source of antimicrobials**. Participants mention a strong dependence on AHPs for administering antimicrobials to their cattle: "*For the case of treatment, the doctor is the one who knows the [antimicrobials].*" (FGD Mukono), and "*I just called the Sub County Veterinarian. This is because I do not know the drugs to use in treating" (FGD Wakiso)*, and "*It is our doctors who treat animals. We do not have any drugs at home. He comes with it to treat the animal*." (FGD Mukono).

**Limited access to AHP services is a barrier to following complete treatment.** One respondent said that dosages are often not thoroughly followed due to the lack of availability and accessibility of AHP services. *"Dosage is not fully followed because doctors [AHPs] sometimes have other programs and also sometimes take long to arrive."* (FGD Wakiso).

**Lack of communication and aftercare concerns with AHPs**. A recurring theme was concerned with communication and aftercare practices provided by some AHPs, as study participants felt they often focused on treatment alone, neglecting to provide comprehensive guidance. Quotes highlighted a lack of clear explanations and follow-ups: "*Sometimes the veterinary doctors do not explain to you the diseases they just treat and go and do not even take the initiative to follow up.*" Another respondent agreed, "*When the doctor comes back, he can give you advice, but most of them do not come back to check on the progress.*" (FGD Wakiso).

Table 5. Composition and contextual characteristics of focus group discussion (FGD) groups.

| FGD No, | District/ Sub-county | Participants (gender) | Production system | Typical herd size (range) | Other livestock |
|---|---|---|---|---|---|
| 1 | Mukono District (Naluvule & Makata villages) | 6 (5 men, 1 woman) | Mix of zero-grazing, tethering, and free-range; mostly cross-bred cattle | 1–5 cows | Goats, pigs, poultry |
| 2 | Mukono District (Nakisunga Sub-county) | 8–10 (mixed sex) | Mainly tethering on communal land; some semi-intensive or zero-grazing | 1–10 cows | Goats, pigs, poultry, sheep |
| 3 | Wakiso District (Gayaza Sub-county) | 4 (2 men, 2 women) | Combination of zero-grazing and free-range; seasonal alternation between systems | 1–3 cows | Goats, pigs, poultry, ducks |

**AHPs address diseases reactively rather than preventively by AHPs**: Participants felt AHPs primarily focused on disease treatment during visits, neglecting to provide advice on disease prevention strategies. "*The vets, though some do not offer comprehensive information, only come when an animal is sick.*" and "*[I do not follow biosecurity measures], and also Vets do not say anything about [biosecurity]*" (FGD Wakiso).

**Perceived financial focus of AHPs**. Additionally, a few participants believed that some AHPs prioritise economic gain over animal well-being. "*Most doctors are money-oriented.*" (FGD Wakiso).

## Discussion

In this study with small to medium dairy cattle farmers in central Uganda, three key insights regarding antimicrobial use practices were identified. Consistent with our objective to identify factors related to prudent antimicrobial use practices we found that that first, prudent use was significantly associated with access to professional veterinary services, negatively associated with reported levels of diseases to a lesser extent, beliefs on issues related to prudent use (withdrawal, antibiotics as prevention and cessation of treatment early). Second, we found that animal health seeking practices were related to satisfaction with the professional sector, patterns of reliance on the professional versus information sector, husbandry practices and disease levels. In addition, there were significant geographic differences in animal health seeking practices. Lastly, our findings indicate that self-reported AMU relevant practices did not always align with knowledge and beliefs related to AMR given that only one knowledge indicator (withdrawal) and two belief questions were selected through LASSO regressions, which included seven different indicators of knowledge and beliefs.

Below, we integrate the quantitative and qualitative findings to situate results within broader socio-cultural and psychological contexts and propose directions for future research.

### Animal health seeking and inappropriate antimicrobial use

Our analysis indicates that accessibility to and positive attitudes toward animal health professionals are positively associated with more prudent antimicrobial use and related practices while reliance on informal sources is negatively related. This is consistent with previous work in Uganda [25,26] as well as globally [27–30]. Consistent with qualitative data, significant differences in animal health seeking outcomes were also documented across locations (sub-countries). In addition, the negative correlation between farmers keeping cattle free-range and utilization of the professional sector and positive association with reporting challenges accessing vets may also reflect the importance of location, as free-range systems are more likely to occur in rural areas where there are often fewer animal health services.

However, not all AHPs are perceived equally. While both government and private veterinarians play an essential role in antimicrobial stewardship, our quantitative and qualitative data combined to demonstrate that farmers heavily rely on private veterinarians for advice and treatment and satisfaction levels with private providers, but not government, positively associated with seeking professional guidance in animal health treatments. In contrast, the data reveals that satisfaction with government veterinarian is negatively impacted accessibility challenges. Discontent with government AHPs indicates a broader historical trend, beginning with structural adjustment programs in the 1980s, where governments were required to substantially decrease funds to government animal health services to secure loans from global lenders such as the World Bank and International Monetary Fund [31–33]. Any service gaps created through these adjustments were expected to be filled by the private sector, but Uganda has not fully bridged the gap. Indeed, this shortfall is visible in the qualitative data, where participants, while accessing private services more often, expressed frustration over difficulties in obtaining guidance or aftercare. Some, not unexpectedly given the drive towards privatization of veterinary care, also perceived veterinarians as prioritising financial gain over animal health and welfare.

Within the context of this study, the prominent influence of private-sector veterinarians among participants in urban and peri-urban areas might have implications for addressing AMR. Currently, mitigation programs are mainly driven by government and intergovernmental organisations, which primarily collaborate with public-sector partners. However, our

results re-emphasise the critical role of private animal health services as key points of contact with farmers. This presents both a challenge and an opportunity. There is scope to consider how existing frameworks and mechanisms in Uganda could be adapted to facilitate closer collaboration with private-sector stakeholders, recognising them as essential partners in promoting responsible antimicrobial use and disease prevention while maintaining oversight, quality assurance, and alignment with national policies.

## Knowledge, awareness and inappropriate antimicrobial use practices

Despite considerable efforts by governments and international organisations to raise knowledge and awareness about AMR in Uganda, our findings indicate that knowledge alone is often not correlated with more prudent antimicrobial use and related practices. Farmers who knew about AMR, withdrawal, and residues, and who held more prudent beliefs on antimicrobial use, did not necessarily report engaging in more responsible use practices. This lack of association underscores a persistent knowledge-action gap, which s aligns with the behavioural science literature emphasising that knowledge is a necessary but insufficient driver of behaviour change [34]. Despite persistent evidence of a knowledge-action gap within knowledge, attitudes, and practices studies of AMU and AMR relevant policies and programs continue to prioritise training and awareness as key interventions [35,36]. Such policies and programs are often founded on the assumption that knowledge deficits are the primary barrier to prudent AMU practices, particularly in LICs and MICs [37], overemphasizing the role of an individual farmer and underestimating the context in which farmers make decisions [31]. This logic is reflected in key policy documents, where training and awareness campaigns are positioned as a central mechanism for addressing AMR. For instance, the Global National Action Plan on AMR, a leading policy framework by the FAO/WOAH/WHO Tripartite Collaboration, prioritises the goal of "improving awareness and understanding of antimicrobial resistance through effective communication, education, and training" [38]. Uganda's AMR-NAP follows a similar structure. These policies often predefine the solution rather than first diagnosing the factors influencing AMR-related behaviours in a specific context and only then tailoring interventions accordingly. Moreover, traditional education models underpinning many AMR awareness campaigns tend to view learners as passive recipients of information, a concept often referred to as the "empty vessel" model of education. However, research in cognitive psychology and learning sciences has long established that knowledge is constructed through experiential learning, contextual relevance, and social interaction [39,40]. Farmers do not simply absorb new information; they integrate it with their existing knowledge structures, including context-specific knowledge of their animals and farming environments, gained through generations of experience.

Instead of knowledge and beliefs concerning AMR, our results suggest that farmers' antimicrobial use and related practices are correlated with experiences engaging the professional and informal veterinary services and that reliance on the professional sector is usually correlated with prudent practices while reliance on the informal sector is associated with less prudent practices. Models assessing the correlations between animal health seeking and farmer attributes found that those who had training on animal health in the past were more likely to utilize informal sources. This reliance on informal sources finding could be explained if more experience with animal health issues creates false confidence in diagnosing and treating cattle without consulting a professional veterinary service or that the nature or content of the training received did not sufficiently emphasize antimicrobial stewardship. Findings from the FGDs lend some support to the initial interpretation, with farmers justifying self-administration of medication by using antimicrobials that "have worked in the past", suggesting a pattern driven by perceived past successes. Previous research also noted similar tendencies where experienced farmers often consider themselves more knowledgeable than veterinarians and may resist external advice [41,42]. In this context, farmers may overestimate their ability to diagnose and treat livestock diseases, potentially reflecting an overconfidence bias [43]. However, we also document a positive correlation with the number of diseases in dairy herds reported as common by the farmers and non-prudent antimicrobial use practices. Higher disease burden could lead farmers, given the documented accessibility issues to both to engage in more frequent non-prudent practices and make it more likely that they seek out animal health training to cope with these ongoing challenges. In such cases, the observed correlation would

not necessarily imply that the training itself led to misuse, but rather that both training uptake and higher AMU are linked to a challenging disease environment. framing the role of KAP in AMR research

## Rethinking the role of KAPs in AMR research

KAP studies are widely used in AMR policy and research. While KAP studies offer valuable insights, they now often reaffirm well-established findings and can reinforce overly reductionist interpretations of complex decision-making processes and therefore risks falling short in understanding the more complex and context-dependent drivers of AMR-relevant practices. Our findings illustrate this weakness: regression models built from typical KAP variables explained only 7–24% of the variance in antimicrobial use and related practices (pseudo $R^2 = 0.20$). This low to modest explanatory power likely reflects both the limitations of our unvalidated scales and a narrow theoretical scope, where KAP variables are insufficient to capture the complexity of farmers' behaviour and decision-making, which are likely better explained by factors that sit outside the traditional KAP constructs, such as other psychological and cognitive factors, economic considerations, systematic and structural barriers, as well as social and cultural dynamics. Taken together, these findings prompt a reconsideration of how KAP tools are used to study AMR-related behaviours, to overcome theoretical as well as methodological weaknesses.

On the theoretical side, KAP studies should be coupled with social and behaviour science theories. In practice, integrative behaviour change models, such as the Theoretical Domains Framework [44] and the Behaviour Change Wheel [34], may offer a structured approach to identifying factors influencing AMR-related practices and guiding theory-informed behaviour change interventions beyond knowledge and attitudes alone. Recent applications of these frameworks, ranging from veterinary prescribing in the UK [45] and Bangladesh [46] to biosecurity efforts among smallholder poultry farmers in Ghana [47], demonstrate the value of these models in generating actionable, contextually relevant insights. The need for theory-driven behavioural research across sectors is already highlighted in the One Health research agenda on AMR [48]. Yet behaviour change approaches remain largely underexplored, particularly in the veterinary and livestock sectors [49–54]. For instance, a recent review identified 301 behaviour change studies on AMR, but only 11 focused on the animal health sector, compared to 290 on human health [55].

There is also a considerable yet largely untapped potential for interdisciplinary collaboration with researchers in the social and behavioural sciences. Such collaboration needs to extend beyond communication strategies to make fundamental contributions to our understanding of decision-making processes, risk perception, cognitive biases, and habit formation that underpin antimicrobial use. Engagement with anthropology and psychology can illuminate decision processes, risk perception, cognitive biases, habits, and social norms that shape antimicrobial use. For example, an anthropological investigation of food preferences among Maasai pastoralists in Tanzania was used to inform an intervention aimed at reducing the risk of AMR transmission through raw milk consumption [56]. Similarly, psychological research has shown that framing AMR as a "silent pandemic" can unintentionally lower perceived urgency, thereby undermining public engagement in mitigation efforts [57].

Importantly, methodological strengthening is needed so that KAP instruments measure what they intend to measure and are comparable across settings. Collaboration with social and behavioural scientists, particularly experts in psychometrics, can support the development of theory-linked, validated multi-item scales for key constructs. Validated measures reduce measurement error, improve causal modelling, and enable meaningful comparisons across subgroups and over time (e.g., via tests of reliability, construct validity, and measurement invariance). Standardised, psychometrically sound tools would make individual studies more reliable and, critically, would allow accumulation of evidence across contexts to inform policy and stewardship programmes.

## Study limitations

The cross-sectional nature of our study provides only a snapshot of a specific point in time, limiting our ability to establish causality or determine the direction of a relationship. Similarly, the factors identified as correlates of AMU and

AHP-seeking behaviours in our exploratory regression models should be interpreted with caution. Findings should be read as hypothesis-generating that need confirmation in studies with pre-specified conditions to establish robust causal links.

Moreover, our measures are self-reported and therefore susceptible to recall error and social-desirability bias: respondents may under-report practices perceived as undesirable and over-report recommended practices [58,59], resulting in conservative estimates of antimicrobial misuse. The knowledge–action gap observed in the data is thus likely more pronounced in routine practice.

Our survey used convenience recruitment, which improves feasibility but provides unknown selection probabilities. Our sample also included populations from urban and peri-urban areas, which likely over-represents farmers with better access to information and animal health services. Our sample includes a high proportion of men with relatively high educational levels. While this profile is typical of per-urban and urban smallholder dairy ownership [17,24], this limits the generalizability of your results, particularly to rural populations and women-led dairy operations. Accordingly, our study should be interpreted as sector-representative of peri-urban/urban smallholder dairy systems in the sampled districts rather than nationally representative. However, even within a comparatively well-connected and formally educated cohort, our findings still revealed widespread suboptimal practices, suggesting that barriers extend beyond information and access and may be at least as pronounced among harder-to-reach farmers.

A further limitation is the potential for respondent fatigue. The survey completion time varied significantly, ranging from 45 to 90 minutes. This wide variation, driven by differences in farmer engagement and farm complexity, introduces the possibility that participants in longer interviews may have experienced fatigue. This could have affected the quality and thoughtfulness of their responses, particularly towards the end of the survey.

Potential biases were partly mitigated using qualitative findings from FGDs, which provided contextual insights into reported behaviours. However, we recognise that three FGDs are insufficient to fully capture complex behavioural phenomena. Our qualitative component was explicitly designed to contextualize the survey instrument and therefore we did not aim to, and do not claim to, have reached data saturation. Although qualitative methods literature notes that saturation can sometimes be observed within 3–4 FGDs when participants share similar experiences [60] the themes we report should be interpreted as indicative rather than comprehensive. Group settings may also inhibit disclosure of sensitive practices.

Finally, a primary limitation, common in this literature and applicable to our study, is the use of psychometrically unvalidated KAP instruments. Without evidence of internal consistency, and construct validity, we cannot be confident to measure the intended latent psychological constructs of "knowledge" or "attitudes". In such cases, forming composite scores can mix domains, inflate measurement error, and weaken observed associations in regression models. We therefore interpreted indicators cautiously and, where appropriate, at the item level, rather than relying on composite scores.

## Future direction

Beyond the KAP considerations noted above, future work should directly address our study's methodological constraints to strengthen inference and policy relevance.

First, designs that permit causal interpretation are needed. (Quasi-)experimental approaches, and longitudinal cohort studies that follow farms across seasons would allow estimation of change and mediation (e.g., through norms, incentives, or access constraints).

Second, to generalise beyond peri-urban districts, future surveys should replace convenience recruitment with probability sampling and pre-specified stratification (e.g., by sex of the primary decision-maker, production system, and extension access. Similarly, given the male-dominated nature of the sector, gender focused studies are needed to understand antimicrobial use, and animal health seeking patterns and barriers of women-led operations.

Third, as already discussed, measurement needs to move beyond unvalidated KAP items. Where feasible, incorporate objective or low-bias measures, like medicine-cabinet inventories, purchase or treatment records, on-farm audits, and

photo documentation, and use techniques that mitigate social desirability (e.g., indirect questioning, vignettes, or self-administered modules).

Finally, qualitative inquiry should be planned as a core component, particularly in understudied settings. An a priori mixed-methods design (explanatory or exploratory sequential) combining FGDs with in-depth individual interviews and ethnographic approaches (e.g., farm walkthroughs, shadowing) can surface sensitive practices and contextual mechanisms that surveys might miss and enable triangulation.

## Conclusion

This study highlights that farm-level factors, including economic reliance on livestock and limited access to veterinary services, are associated with suboptimal antimicrobial use among Ugandan dairy farmers. Awareness and knowledge of antimicrobial resistance did not translate to better practices. Therefore, addressing AMR will require a shift toward integrating more behaviour-centred approaches underpinned by established behaviour change models, coupled with interdisciplinary collaboration. Our results also indicate a need for, consideration to be given to further fostering partnerships with private veterinary providers, as this study confirms their essential role as key stakeholders in reaching farmers and influencing the effectiveness of national AMR strategies.

### Reflexivity statement

This study was conceptualised to address Uganda's priority research and policy needs outlined in the National Action Plan on AMR. It focused on smallholder dairy farmers, a critical gap in livestock sector research. Local researchers and practitioners from Makerere University and the FAO were integral to the study design, ensuring that the study addressed contextually relevant themes.

Funding was allocated to support the local research team through training and logistical support for fieldwork, enabling them to lead data collection. Their contributions are explicitly acknowledged through authorship and acknowledgements. Research team members were provided access to study data and were actively involved in interpreting the findings through online meetings, ensuring that the analyses were locally relevant.

Capacity building involved hands-on training in mixed methods, survey design, and data analysis, led by AB and MAC during a week-long visit to Uganda. Early-career researchers (at the master's and PhD levels) serving as enumerators were included in the authorship to foster their professional growth, with ongoing support for the development of statistical and qualitative skills.

Findings will be disseminated through knowledge-sharing sessions and co-designed interventions, ensuring accessibility for stakeholders, including participants.

Safeguarding procedures included obtaining ethical approvals, informed consent, and implementing measures to ensure confidentiality and safety during data collection.

## Supporting information

**S1 Annex A. Descriptive statistics, LASSO results and model diagnostics.**
(DOCX)

**S2 Annex B. Power analysis.**
(DOCX)

**S3 Annex C. Qualitative codebook.**
(DOCX)

**S4 Annex D. KAP instrument.**
(DOCX)

**S5 Annex E. Data.**
(XLS)

## Acknowledgments

First and foremost, we would like to extend our gratitude to the participants in this study for sharing their insights and experiences, and for warmly welcoming us into their farms and homes. We would also like to thank Rose Kibanya for logistical support, without which this research would not have been possible. Our sincere thanks also go to our dedicated team and co-authors at Makerere University in Uganda (alphabetically): James Muleme, James Natweta Baguma, Lordrick Alinaitwe, Methodius Tubihemukama, Rogers Musiitwa, and Simon Peter Musinguzi. We thank Markus Lipp, Charles Bebay, Willington Bessong Ojong, and Antonio Querido for their help in implementing this research project. The views expressed here are those of the authors and do not necessarily reflect the views or policies of the Food and Agriculture Organization of the United Nations.

## Author contributions

**Conceptualization:** Anica Buckel, Clovice Kankya, Mark A. Caudell, Tabitha Kimani, Alice Namatovu, Emmanuel Kabali, Jorge Pinto Ferreira, Jeffrey LeJeune.

**Data curation:** Anica Buckel, Clovice Kankya, Mark A. Caudell.

**Formal analysis:** Anica Buckel, Mark A. Caudell.

**Funding acquisition:** Jorge Pinto Ferreira, Jeffrey LeJeune.

**Investigation:** Anica Buckel, Clovice Kankya, Mark A. Caudell, Alice Namatovu, Lordrick Alinaitwe, James Natweta Baguma, Rogers Musiitwa, Methodius Tubihemukama.

**Methodology:** Anica Buckel, Clovice Kankya, Mark A. Caudell.

**Project administration:** Clovice Kankya, Tabitha Kimani, Alice Namatovu, Jorge Pinto Ferreira, Jeffrey LeJeune.

**Resources:** Jeffrey LeJeune.

**Supervision:** Anica Buckel, Clovice Kankya, Tabitha Kimani, Alice Namatovu, Junxia Song, Jorge Pinto Ferreira, Jeffrey LeJeune.

**Validation:** Alice Namatovu, Emmanuel Kabali, Jorge Pinto Ferreira, Jeffrey LeJeune.

**Writing – original draft:** Anica Buckel, Clovice Kankya, Mark A. Caudell.

**Writing – review & editing:** Tabitha Kimani, Alice Namatovu, Lordrick Alinaitwe, James Natweta Baguma, Rogers Musiitwa, Methodius Tubihemukama, Junxia Song, Emmanuel Kabali, Jorge Pinto Ferreira, Jeffrey LeJeune.

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
