## [Decision Letter · Decision Letter 0]

16 Sep 2025

Dear Dr. Buckel,

Thank you for submitting your manuscript to PLOS ONE. After careful consideration, we feel that it has merit but does not fully meet PLOS ONE’s publication criteria as it currently stands. Therefore, we invite you to submit a revised version of the manuscript that addresses the points raised during the review process.

We look forward to receiving your revised manuscript.

Kind regards,

Chisoni Mumba

Academic Editor

PLOS ONE

Journal Requirements:

2. Please amend the manuscript submission data (via Edit Submission) to include author Methodius Tubihemukama

3. Please amend your authorship list in your manuscript file to include author Methodius Tubihemukam

4. Please ensure that you refer to Figure 1 in your text as, if accepted, production will need this reference to link the reader to the figure.

5. We note that Figure 1 in your submission contain map images which may be copyrighted. All PLOS content is published under the Creative Commons Attribution License (CC BY 4.0), which means that the manuscript, images, and Supporting Information files will be freely available online, and any third party is permitted to access, download, copy, distribute, and use these materials in any way, even commercially, with proper attribution. For these reasons, we cannot publish previously copyrighted maps or satellite images created using proprietary data, such as Google software (Google Maps, Street View, and Earth). For more information, see our copyright guidelines: http://journals.plos.org/plosone/s/licenses-and-copyright.

6. We note you have included a table to which you do not refer in the text of your manuscript. Please ensure that you refer to Tables 3, 4, and 5 in your text; if accepted, production will need this reference to link the reader to the Tables.

Additional Editor Comments:

Please attend to all the comments from the reviewers. Pay attention to the attached documents (comments) from Reviewer 1 and 2.

Reviewers' comments:

Reviewer's Responses to Questions

**Comments to the Author**

1. Is the manuscript technically sound, and do the data support the conclusions?

Reviewer #1: Yes

Reviewer #2: Yes

Reviewer #3: Yes

2. Has the statistical analysis been performed appropriately and rigorously?

Reviewer #1: No

Reviewer #2: Yes

Reviewer #3: Yes

3. Have the authors made all data underlying the findings in their manuscript fully available?

Reviewer #1: No

Reviewer #2: Yes

Reviewer #3: Yes

4. Is the manuscript presented in an intelligible fashion and written in standard English?

Reviewer #1: Yes

Reviewer #2: Yes

Reviewer #3: Yes

Reviewer #1: This manuscript holds good promise but will benefit from more rigour especially the methodology and results. Some of the results in table 2 for example are not correctly calculated. Also, some methods were not clearly justified. For example, there was no justification for the said sample size.

Reviewer #2: The manuscript is technically sound, with appropriate methodology, statistical analyses, and data that support the conclusions. It is clearly written in standard English and accessible to the intended audience.

Some improvements are recommended to strengthen clarity and transparency. In the methodology, please provide more detail on the study design, sample size estimation, sampling strategy, and definition of the sampling unit. Clarify the administration of questionnaires, selection of FGD participants, and how potential bias was addressed. Distinguish clearly between qualitative and quantitative methods, and specify the statistical software and stepwise approach to model development.

In the results, consider adding information on internal validity or reliability measures, as well as how model goodness-of-fit and performance were assessed.

Overall, this is a strong manuscript, and addressing these points will enhance its clarity and rigor.

Reviewer #3: - On the technical soundness of the manuscript, there is only one concern. This is because the quantitative part was done with small holder farmers but for the quantitative part, it was not restricted to small holder farmers only. What proportion did the small holder farmers form in the groups. There is a risk of the farmers with more animals being more vocal in such groups due to the confidence they have and others may feel intimidated as they are 'smaller' farmers. Looking at eh sample size, there was no shortage of small holder farmers and it is not clear what advantage there was in bringing in the 'bigger' farmers. The medium sized dairy farm is also not defined in relation to the number of animals, as the small sized farm has been described.

In relation to the statistics, the methodology section inciates a p-value cut off of 0.1, but in the reporting of results in the tables, a variety of cut-offs are indicated. Consider putting he actual p-value for each of the variables tested in the table itself for ease of understanding. Alternatively, consider indicating what was significant or not based on the selected p-value.

**Do you want your identity to be public for this peer review?** For information about this choice, including consent withdrawal, please see our Privacy Policy

Reviewer #1: No

Reviewer #2: No

Reviewer #3: No

---

## [Author Response · Author response to Decision Letter 1]

30 Nov 2025

We would like to sincerely thank the editors and both reviewers for taking the time to carefully reading of our manuscript and for providing detailed, constructive feedback. We greatly appreciate the thoughtful comments, which have helped us to substantially strengthen the clarity, methodological transparency, and overall quality of the paper.

In revising the manuscript, we have addressed each point raised by the reviewers in detail and implemented corresponding changes throughout the text. Major revisions include a clearer articulation of the study’s design and analytical approach, the replacement of stepwise regression and composite scales with a LASSO–logistic modeling framework, expanded methodological detail on power analysis, inter-rater reliability, and model diagnostics, and a more explicit discussion of study limitations and future research directions.

We believe these revisions have improved both the scientific rigour and readability of the manuscript.

Detailed responses to each reviewer’s comments are provided in the tables below.

Reviewer 1

Inconsistent terminology and causal

language where correlational terms would be appropriate. Key findings are buried within methodological details rather than clearly highlighted This language has been corrected throughout the manuscript. While we were careful to point out the correlational nature of our findings in the Results Section the Discussion did frame results in causative terms.

In addition, we now trace all key findings from Results (both quantitative and qualitative) to the Discussion

Generally clear but lack important methodological details such as reliability coefficients for constructed scales and detailed missing data patterns Given the lack of validated scales within areas relevant to AMR in smallholder livestock keeping communities (i.e., KAP and animal health seeking practices), we used a linear sum of several variables representing antimicrobial use practices and animal health seeking practices. However, upon reflection of the reviewers’ comments we realized that this approach (i.e., linear summed scale) was not the most informative in identifying correlates of non-prudent practices and animal health seeking practices. Consequently, we dispensed with the use of scales and instead used a LASSO regression approach where all variables contained within the scales were included in the variable selection process. Variables selected through 10 fold cross validation were then entered into logistic models whose outcomes were the same as the original manuscript. In addition, we acknowledge the issue of a lack of validated scales in the Study Limitations

We also provide more discussion of the patterns of missing data in Lines 345 - 349.

No power analysis, sample size justification, inter-rater reliability for qualitative coding, or comprehensive missing data analysis provided We agree with the reviewer that all of these are necessary for the manuscript. We have provided power analysis justification in Lines 170, and 257 - 266, inter-rate reliability for qualitative coding in Lines Lines 391 - 398, and missing data explanation in Lines 345 - 349.

"We found that (1) farmers' AMU practices are shaped by attitudes toward veterinarians..." - The cross-sectional design cannot establish causality, yet the language implies causal relationships throughout. This should be "farmers' AMU practices were associated with attitudes toward veterinarians..."

"knowledge about AMR did not predict better practices" - This overstates findings from exploratory regression models that explain limited variance (R² = 0.18-0.20 for model 2 and 1 respectively).

This part of the abstract was rephrased to avoid overstatements.

Line 19: The definition of metaphylaxis is incomplete and potentially misleading. They define it as "clinically healthy animals sharing premises with symptomatic animals," which is more of a description of the situation rather than the actual antimicrobial intervention. The accepted veterinary definition of metaphylaxis is the treatment of clinically healthy animals that have been exposed to infectious agents, typically when some animals in the group are already showing clinical signs of disease. It's a targeted group treatment strategy.

This was corrected to “the treatment of clinically healthy animals in a group that have been exposed to an infectious agent, following the diagnosis of the disease in other animals within the same group”

Line 15-17

: The repeated use of "LICs and LMICs" throughout the manuscript reflects a misunderstanding of standard World Bank classification terminology. "LMICs" (low- and middle-income countries) already includes low-income countries as a collective term for all non-high-income countries. The redundant phrasing "LICs and LMICs" should be replaced with simply "LMICs" or "low- and middle-income countries (LMICs)" on first usage. Thank you for noticing this mistake. We agree the phrase “LICs and LMICs” is incorrect and appreciate the opportunity to clarify our terminology choices. The specific wording “LICs and LMICs” in our draft was an unintended search-and-replace artifact when we corrected earlier instances of “LMICs.” We have fixed this throughout.

Our intention is to align with the World Bank’s official income groups. Because the WBG does not define a single combined group called “LMICs,” we opted to use the two official groupings together (low-income countries (LICs) and middle-income countries (MICs)) where relevant. (We recognise that “LMICs” is standard shorthand in the research community. Our choice is simply to mirror the World Bank taxonomy we cite.)

The comparison between Uganda and high-income countries lacks nuance about different production systems and contexts that might explain these differences beyond just veterinarian ratios.

We added more context for nuance in the comparison:

line 28-35

The claim about "absence of any prior KAP study" in Uganda's livestock sector appears overstated without comprehensive literature search evidence. Consider qualifying this statement. For example, here is a quick search result; Kahunde M. A.Odoch T.Owiny D. O.Kankya C.Kaelin M. B.Hartnack S. (2023). Knowledge, attitudes and practices towards antibiotic use and resistance in Kyegegwa district, Uganda – a questionnaire study. medRxiv. 10.1101/2023.04.06.23288253; https://www.researchgate.net/publication/389052753_Smallholder_farmers'_knowledge_attitud e_and_practices_towards_use_and_disposal_of_veterinary_antibiotics_in_Kassanda_District_Uga nda_-_A_cross_sectional_survey

Thank you very much. We agree. The language has been adjusted.

Line 109-114

"failure, and even the emergence of "Multidrug-resistant bacteria" that render no drugs" - This phrase is awkward and unclear. Should be rephrased for clarity.

We agree. This phrase has been removed entirely

The convenience sampling approach through veterinary extension workers introduces systematic selection bias not adequately addressed. Farmers connected to extension services may differ in knowledge, practices, and veterinary access from the broader population We agree with the reviewer and have expanded our discussion concerning sampling biases in the limitation and future direction section.

line 1131-1142

We also have clarified the sampling approach. We worked through the district veterinary office rather than private vet/extension services), who provided a list of registered farmers in the study district. We recruited via convenience sampling from this list. Government extension workers just sometimes supported enumerators in finding the location.

Line 174-181

The definition of "smallholder" as 1-15 cattle appears arbitrary without supporting rationale from literature or local context. What constitutes "smallholder" varies significantly across regions and should be justified We agree. We now state our herd-size band as an operational definition for this study and justify it with literature.

Line 161-169

Only three focus group discussions are insufficient for understanding complex behavioural phenomena. No discussion of data saturation is provided, and this limitation significantly undermines the qualitative component's validity. We agree with the reviewer. In the original submission we only briefly noted the limited number of FGDs.

We have now elaborated the paragrpah in the limitations section to explicitly acknowledge that three FGDs are insufficient for fully understanding complex behaviours, that we did not aim to (and do not claim to) have reached data saturation. Line 1148-1156

We also clarify the rationale for conducting the FGDs, which was previously just metioned within the Public Involvement section in the methods section (line 204-205)

Survey duration of 45-90 minutes suggests potential respondent fatigue that could affect data quality, yet this substantial variation is not addressed or explained.

We have addressed this and explained the variation. Line 220-221

Multiple statistical methodology concerns: Stepwise regression with p=0.1 cutoff increases Type I error risk and is problematic for exploratory analysis. The creation of composite scales without psychometric validation (reliability, construct validity) is concerning. The "antimicrobial misuse scale" combines conceptually different behaviours without theoretical justification for equal weighting.

We agree with the reviewer that better analytical approaches exist in examining correlates of AMU and relevant practices, particularly given the lack of validated scales. We now use a LASSO variable selection approach in place of stepwise regression, allowing us to dispense with scales and specify standard logistic models with no stepwise selection.

"Regression diagnostics were run for all models, generally indicating good model fit (see Annex A)" - No Annex A is provided. Either include diagnostic results or remove this reference.

Regression Diagnostics are now provided for the logistic models specified after variable selection through LASSO in the Supplement S1 beginning on page 15-28

Listwise deletion of missing data without analysis of missingness patterns is problematic, especially given the exploratory nature and potential systematic differences in missing responses.

Missing data in the new models is minimal (12 observations from 416) and we explain why these 12 observations were dropped in Lines 345 - 349.

Variable definitions lack sufficient operational detail for replication. For example, how exactly was the "antimicrobial misuse scale" calculated and validated? All scales have now been removed from the analysis, both dependent and independent.

Demographic distribution (60% male, high education levels) suggests potential sampling bias, but no comparison with broader population characteristics is provided to assess representativeness.

We agree that women make up a larger share of the general Ugandan population. However, the sample profile in our study is consistent with this segment of the sector in the study districts where dairy ownership and day-to-day caretaking are predominantly male. The relatively high education levels in our sample reflect the peri-urban/urban context where dairy herds are often side investments of middle-income owners with primary employment outside agriculture.

We have revised the Results (line 424-428) and Limitations section (line 1243 - 1247) to qualify generalisability and avoid population-level claims.

Several percentage calculations in Table 2 are incorrect. For example, 'Owner treats: 25.8%, 98 obs' should be 23.5% (98÷417), 'Prescription obtained never/rarely: 57.5%, 217 obs' should be 52.0% (217÷417), and similar errors exist for peer and worker treatment percentages. These calculation errors require correction throughout the table Table 2 has been removed from the manuscript as it was describing the scales. Detailed descriptive statistics of these variables, as well as all other variables included in the LASSO regression, are now provided in Table X in the Supplement while general statistics are described in the text of the Results Section

Statistical interpretation problems. "Age was negatively related to misuse" implies causation from correlation. The coefficient of -0.13 may lack practical significance despite statistical significance.

We have updated language with the results to emphasize our models assess correlation not causation.

The models explain limited variance (R² = 0.18-0.20), indicating that most factors influencing antimicrobial use remain unidentified. This limitation undermines claims about identifying key correlates.

References to identifying “key correlates” have been removed and we further discuss this issue in line 956-1018

While qualitative insights are valuable, the limited sample (3 FGDs) undermines claims about thematic saturation. Quotes lack sufficient context about speaker characteristics and farm contexts.

See our response to the related comment above.

Briefly, we agree that three FGDs are insufficient to claim saturation. The qualitative component was designed as exploratory triangulation rather than a stand-alone saturated analysis. We have (i) removed any implication that saturation was achieved, (ii) expanded the Limitations to acknowledge the small number of FGDs, (iii) included a brief table summarizing each FGD’s composition. (FGD transcripts were anonymized at source and did not include individual identifiers. Quotations are therefore attributed at the group level rather than to specific participants.) Table 5

The authors make policy recommendations about private veterinarians without acknowledging how their methodological limitations (convenience sampling, cross-sectional design) affect the generalisability of these recommendations. The Rwanda comparison lacks critical analysis of contextual differences.

To address it, we removed the Rwanda comparison to avoid overextending the discussion beyond the scope of our data and to maintain focus on Uganda. Our original intent was to illustrate an example of regional public–private collaboration, however, we agree that the contextual and structural differences between Uganda and Rwanda warrant a more nuanced discussion than our data or the scope of this publication could support.

The future research directions don't address their own study's methodological shortcomings. No mention of needs for probability sampling, longitudinal designs, scale validation, etc. They focus on general research directions rather than building on lessons from their study's limitations.

The limitations section significantly understates problems. While they mention cross-sectional design and convenience sampling, they completely omit:

• Measurement validity of composite scales (not mentioned at all)

• Detailed discussion of selection bias implications

• Generalisability constraints (only briefly mentioned)

• More thorough discussion of social desirability bias impacts

We have made substational changes to both the limitation and future direction section throughout.

General research directions, particularly concerning KAP and interdisciplinary research are now seperated from study specific limitations (line 961 - 1064)

The conclusion uses causal language ("influence suboptimal antimicrobial use") when only correlational relationships were established, and makes broad policy recommendations despite methodological limitations.

We have deleted all references to causal language throughout the Results and Discussion and provide a more nuanced discussion of policy recommendations.

Reviewer 2

The reader might find it difficult to understand the statement in lines 3-5. Provide more context either by citing sources or directly linking it to the main objective. We revised the paragraph to better link AMR to the SDGs and added a bridging statement connecting it to the study’s objectives.

Line 4-7

Even though it may be implied in line 44, it is not clear where Uganda falls (economic classification)

Line 36-37

Check citation in line 49

All citations in that section were checked and corrected to align with reference numbering and formatting.

Line 59. The sentence seems to be missing a conne

---

## [Decision Letter · Decision Letter 1]

14 Dec 2025

Minding the Knowledge-Action Gap: 

Results from a Mixed-Methods Study of Antimicrobial Use Among Dairy Farmers in Central Uganda

PONE-D-25-42552R1

Dear Dr. Buckel,

We’re pleased to inform you that your manuscript has been judged scientifically suitable for publication and will be formally accepted for publication once it meets all outstanding technical requirements.

Kind regards,

Chisoni Mumba

Academic Editor

PLOS One

Additional Editor Comments (optional):

Reviewers' comments:

Reviewer's Responses to Questions

**Comments to the Author**

Reviewer #1: All comments have been addressed

2. Is the manuscript technically sound, and do the data support the conclusions?

Reviewer #1: Yes

3. Has the statistical analysis been performed appropriately and rigorously?

Reviewer #1: Yes

4. Have the authors made all data underlying the findings in their manuscript fully available?

Reviewer #1: Yes

5. Is the manuscript presented in an intelligible fashion and written in standard English?

Reviewer #1: Yes

Reviewer #1: No further comments as the authors have sufficiently addressed all my concerns. Well done with the great job of addressing all concerns. I look forward to reading this in print.

**Do you want your identity to be public for this peer review?** For information about this choice, including consent withdrawal, please see our Privacy Policy

Reviewer #1: **Yes: ** Sunday Ochonu Ochai

---

## [Editor Report · Acceptance letter]

PONE-D-25-42552R1

PLOS One

Dear Dr. Buckel,

I'm pleased to inform you that your manuscript has been deemed suitable for publication in PLOS One. Congratulations! Your manuscript is now being handed over to our production team.

Kind regards,

on behalf of

Dr Chisoni Mumba

Academic Editor

PLOS One